# Efficient, narrow-band, and stable electroluminescence from organoboron-nitrogen-carbonyl emitter

Ying-Chun Cheng[1,6], Xun Tang[2,6], Kai Wang ®[1,3] ✉, Xin Xiong[1], Xiao-Chun Fan ®[1], Shulin Luo[1], Rajat Walia[1], Yue Xie[1], Tao Zhang[1], Dandan Zhang[1], Jia Yu[1,4], Xian-Kai Chen ®[1] ✉, Chihaya Adachi ®[2,5] ✉ & Xiao-Hong Zhang ®[1,4] ✉

Organic light-emitting diodes (OLEDs) exploiting simple binary emissive layers (EMLs) blending only emitters and hosts have natural advantages in low-cost commercialization. However, previously reported OLEDs based on binary EMLs hardly simultaneously achieved desired comprehensive performances, e.g., high efficiency, low efficiency roll-off, narrow emission bands, and high operation stability. Here, we report a molecular-design strategy. Such a strategy leads to a fast reverse intersystem crossing rate in our designed emitter $h$-BNCO-1 of $1.79 \times 10^5 \, s^{-1}$. An OLED exploiting a binary EML with $h$-BNCO-1 achieves ultrapure emission, a maximum external quantum efficiency of over 40% and a mild roll-off of 14% at 1000 cd·m$^{-2}$. Moreover, $h$-BNCO-1 also exhibits promising operational stability in an alternative OLED exploiting a compact binary EML (the lifetime reaching 95% of the initial luminance at 1000 cd m$^{-2}$ is ~ 137 h). Here, our work has thus provided a molecular-design strategy for OLEDs with promising comprehensive performance.

Organic light-emitting diode (OLED) technology has shown great potential for applications in next-generation displays and lighting[1,2]. Binary emissive layers (EMLs) composed of only hosts and emitters have natural advantages in low film-fabrication cost, thus making them the mainstream solution in current OLED production lines. However, it is difficult in previously reported OLEDs with binary EMLs to simultaneously show desired comprehensive performances to date, e.g., high color purity, high efficiency, low efficiency roll-off, and long operational lifetime.

To overcome the above long-standing bottleneck, one of the keys is to design novel OLED emitter molecules. Since the seminal work in 2012[3], high-efficiency organic emitters with thermally activated delayed fluorescence (TADF) have attracted much attention from industry and academia, though the color purities of the first-generation TADF materials are not ideal[4,5]. In 2016, Hatakeyama et al. pioneered a rigid fused organoboron–nitrogen (BN) framework[6]. Their distinct BN-based frameworks induce a small energy gap ($\Delta E_{S1T1}$) between the first singlet ($S_1$) and triplet ($T_1$) excited states and thus TADF via the so-called multi-resonance (MR) effect, with substantially suppressed geometrical relaxations and thus a narrow emission bandwidth[7–14]. However, OLEDs based on such BN-based MR emitters showed serious efficiency roll-off and poor device operation stability. One of the main reasons is the slow reverse intersystem crossing (RISC) rate ($k_{RISC}$) from the triplet to singlet excited states, generally within $10^2–10^5 \, s^{-1}$ in BN-based MR emitters[9,12,15]. The urgent task is thus to propose a judicious molecular-design strategy for MR emitters with a faster $k_{RISC}$ (e.g., $>10^5 \, s^{-1}$), narrow emission bandwidths, and high photoluminescence (PL) quantum yields ($\Phi_{PLS}$).

[1]Institute of Functional Nano & Soft Materials (FUNSOM), Joint International Research Laboratory of Carbon-Based Functional Materials and Devices, Soochow University, Suzhou, Jiangsu 215123, PR China. [2]Center for Organic Photonics and Electronics Research (OPERA), Kyushu University, 744 Motooka, Nishi-ku, Fukuoka 819-0395, Japan. [3]Jiangsu Key Laboratory for Carbon-Based Functional Materials & Devices, Soochow University, Suzhou 215123 Jiangsu, PR China. [4]Jiangsu Key Laboratory of Advanced Negative Carbon Technologies, Soochow University, Suzhou 215123, PR China. [5]International Institute for Carbon-Neutral Energy Research (I2CNER), Kyushu University, 744 Motooka, Nishi, Fukuoka 819-0395, Japan. [6]These authors contributed equally: Ying-Chun Cheng, Xun Tang. ✉e-mail: wkai@suda.edu.cn; xkchen@suda.edu.cn; adachi@cstf.kyushu-u.ac.jp; xiaohong_zhang@suda.edu.cn

In the framework of the Fermi golden rule, $k_{RISC}$ depends on the square of spin–orbit coupling (SOC) between $T_1$ and $S_1$ states and the negative exponent of $\Delta E_{S1T1}$[16]. A large SOC and small $\Delta E_{S1T1}$ are important to accelerating the RISC process. However, the conventional BN-based MR emitters can gain small $\Delta E_{S1T1}$ (e.g., <0.1 eV); their SOC values are commonly small or negligible, usually ~0.1 cm⁻¹[12]. Recently, the heavy atoms [e.g., selenium (Se)] were incorporated into BN-based MR emitters to enhance SOC (e.g., up to ~2.8 cm⁻¹) via the heavy-atom effect[17,18], eventually leading to a fast $k_{RISC}$ ($2.0 \times 10^6$ s⁻¹). Although a promising external quantum efficiency (EQE) of ~36.8% in an OLED with a binary EML composed of a bipolar TADF host and Se-integrated MR emitter was obtained, its spectral full width at half maximum (FWHM) was as broad as 48 nm, which is much broader than that of (*ca.* 34 nm) of its parent compound BNCZ (for its chemical structure, see Fig. 1a)[17,19]. The broader FWHMs of the Se-integrated MR emitters are because the larger-size electronic cloud of the Se atom can be more easily deformed with molecular vibrations, thus increasing geometrical relaxation energies[20]. Moreover, OLEDs exploiting the Se-integrated MR emitters show short device lifetimes, e.g., $LT_{80}$ (the lifetime attenuating to 80% of the initial luminance) at an initial brightness of 1000 cd m⁻² ~1 h[17], which could be ascribed to high chemical activity of organic selenides. While the broader FWHM issue can be addressed by moving heavy atoms to a peripheral decoration rather than MR frameworks, the device's operational stability is still not satisfactory[21]. Therefore, the design and development of MR-TADF emitters composed of only Period-2 elements with a fast $k_{RISC}$ is critical.

In the present work, we thus rationally proposed a promising molecular-design strategy, i.e., hybridization of organoboron–nitrogen and carbonyl (named *h*-BNCO) fragments. Via such a strategy, we fused carbonyl groups into the conventional BNCZ framework and designed a proof-of-concept *h*-BNCO molecule, i.e., *h*-BNCO-1 (Fig. 1a). Compared with BNCZ, *h*-BNCO-1 not only reduces $\Delta E_{S1T1}$ and the $T_1$–$T_2$ energy gap but also substantially increases the SOC between the $S_1$ and $T_2$ states, eventually accelerating its $k_{RISC}$ to $1.79 \times 10^5$ s⁻¹. Moreover, a small FWHM is maintained in *h*-BNCO-1. An OLED exploiting a binary EML with *h*-BNCO-1 doped in a bipolar TADF host yielding a high $\Phi_{PL}$ of 99.6% ± 2% not only achieved ultrapure green electroluminescence (EL) with an FWHM of 39 nm, corresponding to CIE coordinates of (0.24, 0.71) but also showed an impressive maximum EQE of 40.1% with a mild efficiency roll-off of 14% at 1000 cd m⁻². Moreover, *h*-BNCO-1 also exhibited promising operational stability in an alternative OLED exploiting a compact binary EML, with device lifetime $LT_{95}$ at an initial brightness of 1000 cd m⁻² reaching 137 h. Such device performances and stability thus represent the state-of-the-art comprehensive OLED performances. Our work has thus provided a promising *h*-BNCO molecular-design strategy for OLEDs exploiting binary EMLs with high efficiency, high color purity, low-efficiency roll-off, and promising operational stability.

## Results

### Theoretical calculations and *h*-BNCO molecular-design strategy

To first provide a fundamental understanding of the electronic structure of our designed *h*-BNCO-1 and the conventional BNCZ, quantum-chemistry calculations were carried out. Previous theoretical investigations have demonstrated that the electron correlation effect is important in MR molecules, and popular DFT approaches cannot provide an exact description of the energies and characteristics of their excited states[22–24]. Here, the high-level Coupled-Cluster quantum-chemistry approach was exploited to examine the electronic-structure properties of the excited states (for details, see the Computational details section).

As shown in Fig. 1, the conventional BNCZ with a small distorted angle (~8.4°) between carbazole (CZ) and DABNA fragments has a $\Delta E_{S1T1}$ of 0.2 eV and a $\Delta E_{T2T1}$ of 0.26 eV. The SOC between the $S_1$ and $T_1$/$T_2$ states is small, ~0.07/0.29 cm⁻¹. The combination of small SOC values, large $\Delta E_{S1T1}$, and large $\Delta E_{T2T1}$ thus leads to a slow $k_{RISC}$ (~$1.57 \times 10^4$ s⁻¹). Compared with BNCZ, *h*-BNCO-1 shows a much more distorted geometry, with a dihedral angle of ~25° between the acridone and DABNA groups. Due to the electron-withdrawing effect of the carbonyl groups and the electronic delocalization on the carbonyl groups, the $S_1$- and $T_1$-state energies both decrease, with a reduced $\Delta E_{S1T1}$ of 0.12 eV. Moreover, $\Delta E_{T2T1}$ decreases to 0.18 eV, which is important to accelerating the RISC process. The decrease in $\Delta E_{T2T1}$ (i.e., also the energy splitting between the triplet states of the two same fragments in *h*-BNCO-1) could be induced by its highly distorted geometry since the electronic communication between the two same fragments is blocked. The $SOC(S_1–T_1)$ is negligible, while the $SOC(S_1-T_2)$ is very large, *ca.* 1.26 cm⁻¹. This difference in their SOC values is mainly due to the difference in the excited-state characteristics since the total angular momentum for the electrons must be conserved in any spin-flip event. The $T_1$ and $S_1$ states both mainly show a similar ππ* excitation character, although on the carbonyl groups the $T_1$ state shows a mixing of ππ* and nπ* (see the region circled by the blue-dashed line in Fig. 1d and Supplementary Fig. 1). In contrast to the $T_1$ state, the ππ* excitation character of the $T_2$ state on the BN main body shows an apparent difference with that of the $S_1$ state, as highlighted by the black-dashed line in Fig. 1d. Moreover, the $T_2$ state on the carbonyls shows only nπ* excitation (see the region circled by the green-dashed line in Fig. 1d), which is different from the ππ* excitation character of the $S_1$ state on the groups. Therefore, such a combination of small $\Delta E_{S1T1}$, small $\Delta E_{T2T1}$, and large $SOC(S_1–T_2)$ gives rise to a fast Boltzmann-averaged $k_{RISC}$ value of ~$1.25 \times 10^5$ s⁻¹.

To further demonstrate the advantage of our proposed *h*-BNCO molecular-design strategy, we replaced the two carbonyls of *h*-BNCO-1 with the electron-withdrawing phenyl-boron groups and thus designed a proof-of-concept BN-MR molecule (named BNBNB) without the carbonyls. The results of our theoretical calculations on BNBNB are also shown in Fig. 1 and the corresponding data are summarized in Supplementary Table 1. Clearly, compared with BNCZ, the introduction of phenyl-boron groups also leads to a smaller $\Delta E_{S1T1}$ (0.08 eV) and $\Delta E_{T2T1}$ (0.18 eV), similar to *h*-BNCO-1. However, its SOC value (0.69 cm⁻¹) between the $S_1$ and $T_2$ states is smaller than that in *h*-BNCO-1 due to the absence of the nπ* excitation in the $T_2$ state. Eventually, the Boltzmann-averaged $k_{RISC}$ value (~$1.70 \times 10^3$ s⁻¹) in BNBNB is two orders of magnitude slower than that in *h*-BNCO-1. In addition, a few examples of nitrogen-carbonyl-conjugated frameworks were reported to show MR-TADF properties. However, their device performances were less ideal, showing broader spectral bands and poor efficiencies with serious efficiency roll-offs[25–34]. In short, our calculation results have thus demonstrated that the proposed *h*-BNCO strategy not only reduces $\Delta E_{S1T1}$ and $\Delta E_{T2T1}$ but also increases the SOC values, eventually substantially accelerating the RISC process.

### Photophysics

UV–vis absorption and PL spectra of BNCZ and *h*-BNCO-1 were recorded in toluene ($1.0 \times 10^{-5}$ M) at room temperature. As shown in Fig. 2a and Supplementary Fig. 2a, BNCZ, and *h*-BNCO-1 both display strong and sharp absorption bands at 458 and 489 nm, respectively. The PL spectrum of BNCZ shows a maximum at 477 nm with an FWHM of 0.13 eV (23 nm); in contrast, that of *h*-BNCO-1 shows an evident bathochromic shift to 516 nm due to the incorporation of carbonyl groups, while importantly, its FWHM is well maintained at 0.13 eV (corresponding to 28 nm). Furthermore, in more solvents with various polarities, such PL trends of both emitters basically remained (see Supplementary Fig. 2b, c).

To further determine the key energy levels of both compounds, we measured their fluorescence and phosphorescence spectra in

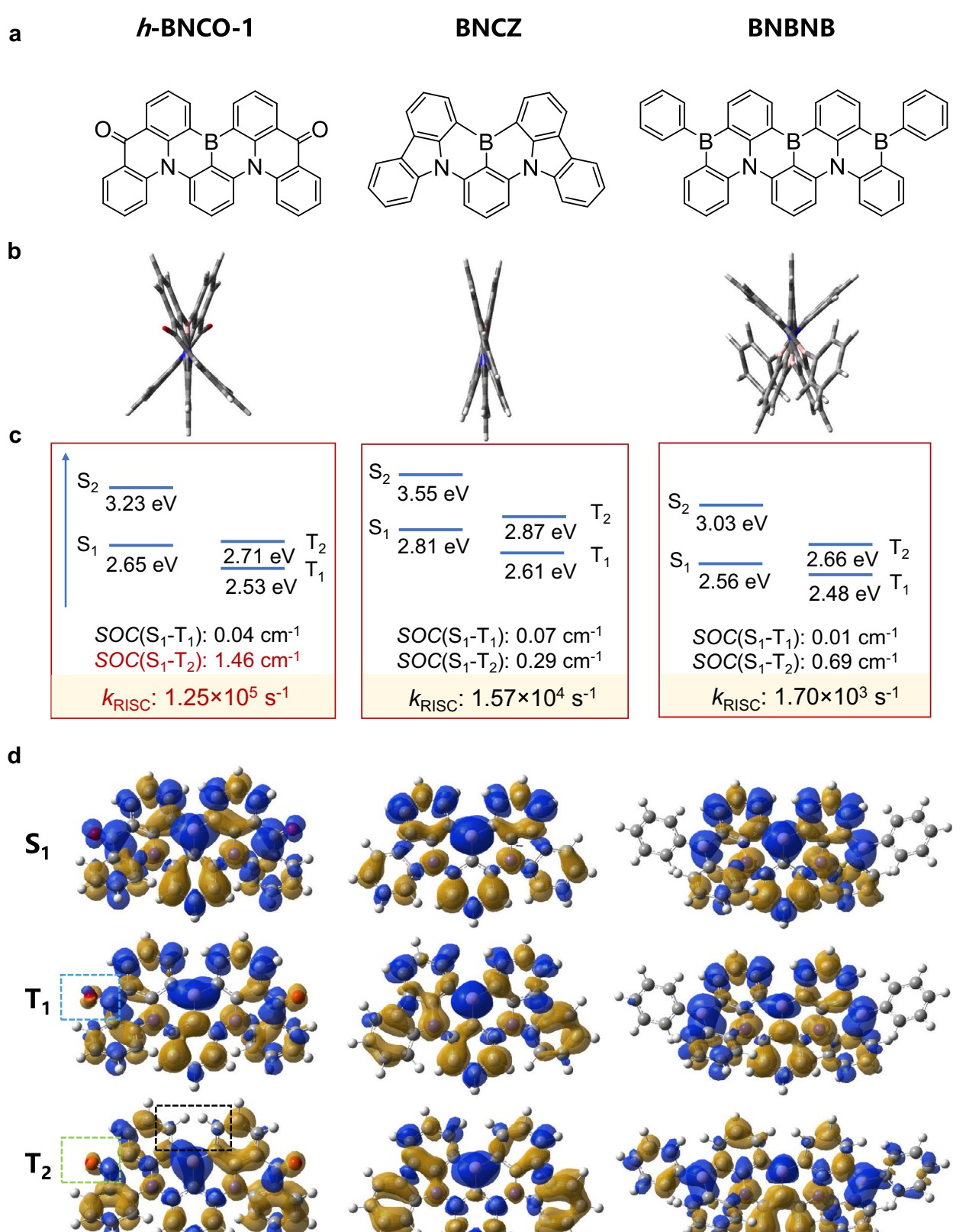

**Fig. 1 | Chemical structures and theoretical calculation results of *h*-BNCO-1, BNCZ, and BNBNB. a** Chemical structures of *h*-BNCO-1, BNCZ, and BNBNB. **b** Optimized geometry structures of their ground (S$_O$) states via the ωB97XD functional with the nonempirically tuned ω value. **c** Calculated excited-state energies and spin–orbit couplings via the high-level STEOM-DLPNO-CCSD method. The estimated $k_{RISC}$ rates are also shown (for details, see the computational details section). **d** Calculated difference-density plots of the S$_1$, T$_1$, and T$_2$ excited states via the high-level STEOM-DLPNO-CCSD method.

**a**

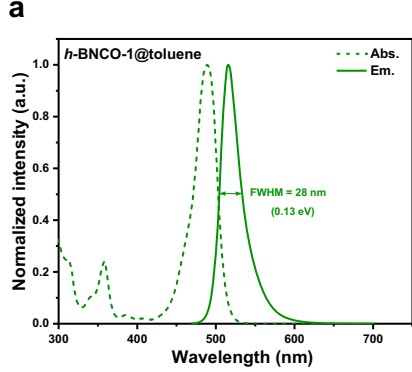

**b**

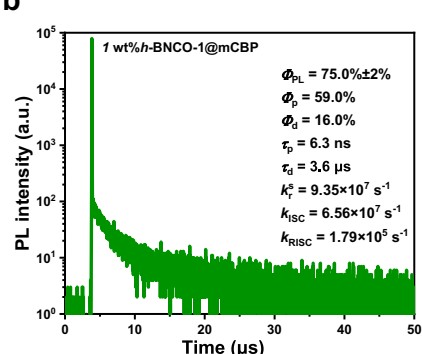

**Fig. 2 | Photophysical properties of *h*-BNCO-1. a** UV–vis absorption and emission spectra of *h*-BNCO-1 in toluene ($1 \times 10^{-5}$ M) at room temperature. **b** Transient PL decays of 1 wt% *h*-BNCO-1 doped in mCBP films at room temperature ($\Phi_{PL}$: PLQY;

$\Phi_p$: prompt fluorescence quantum efficiency; $\Phi_d$: delayed fluorescence quantum efficiency; $\tau_p$: prompt lifetime; $\tau_d$: delayed lifetime; $k_r^s$: radiation rate of singlets; $k_{ISC}$: intersystem crossing rate; $k_{RISC}$: reverse intersystem crossing rate).

frozen toluene at 77 K (shown in Supplementary Fig. 3 and the data are summarized in Table 1). Based on the fluorescence and phosphorescence maxima of BNCZ and *h*-BNCO-1, the energies of their $S_1$ states are estimated to be 2.57 and 2.39 eV, and those of their $T_1$ states are estimated to be 2.46 and 2.36 eV, respectively. Their $\Delta E_{S1T1}$ values were further computed to be 0.11 and 0.03 eV, respectively. This indicates that the introduction of carbonyl groups substantially reduces $\Delta E_{S1T1}$, thus benefitting the RISC process, which is consistent with our theoretical calculation results.

We then prepared a film of 1 wt% *h*-BNCO-1 doped in a non-TADF material 3,3-di(9H-carbazol-9-yl)biphenyl (mCBP). The $\Phi_{PL}$ of the *h*-BNCO-1-based film was evaluated to be 75.0% ± 2%. Figure 2b and Supplementary Fig. 4 depict its transient PL decays at room temperature. An obvious delayed component is detected due to the TADF process. We further evaluated photophysical dynamics based on these data (summarized in Fig. 2b). Importantly, the $k_{RISC}$ of *h*-BNCO-1 reaches an impressive value of $1.79 \times 10^5$ s$^{-1}$, which is much faster than that of BNCZ in the same condition ($1.24 \times 10^4$ s$^{-1}$)[35]. The substantially accelerated RISC can be well explained by our theoretical estimation results. The higher-lying $T_2$ state in *h*-BNCO-1 shows a larger SOC with the $S_1$ state due to carbonyl incorporation, which contributes to the overall RISC process. For OLEDs, the substantially enhanced RISC process can not only enhance triplet utilization but also reduce triplet-polaron/triplet annihilation at high exciton densities, which can give rise to high peak efficiency, suppressed efficiency roll-off at high current densities as well as operational stability.

## OLED performances and stability

An OLED with the binary EML of 1 wt% *h*-BNCO-1 doped in conventional p-type non-TADF host mCBP demonstrated decent device performance (see Supplementary Fig. 5). Here to fully tap the potential of *h*-BNCO-1 and compare their respective electroluminescence (EL) performances, OLED devices with a binary EML composed of only host and emitters were fabricated by employing BNCZ / *h*-BNCO-1 as

emitters, with an optimized device configuration of ITO (indium tin oxide)/HAT-CN (1,4,5,8,9,11-hexaazatriphenylene hexacarbonitrile, 7 nm)/TAPC (N,N-bis(p-tolyl)aniline, 30 nm)/TCTA (tris(4-carbazoyl-9-ylphenyl)amine, 10 nm)/mCBP (10 nm)/ 5-(3-(4,6-diphenyl-1,3,5-triazin-2-yl)phenyl)-7,7-dimethyl-5,7-dihydroindeno[2,1-b]carbazole (DMIC-TRZ): 1 wt% emitter (20 nm)/TmPyPB (3,3'-[5'-[3-(3-pyridinyl)phenyl] [1,1':3',1''-terphenyl]3,3''diyl]bispyridine, 40 nm)/LiF (1 nm)/Al. A bipolar TADF material DMIC-TRZ is selected to be the host matrix, in order to balance the carrier mobilities and improve the $\Phi_{PL}$ of *h*-BNCO-1 to 99.6% ± 2%. The corresponding energy level diagram of the devices is displayed in Fig. 3a.

The device performances are summarized in Table 2 and shown in Supplementary Fig. 6. As shown in Fig. 3b, the BNCZ-based device shows a sky-blue EL spectrum with a peak at 484 nm and a FWHM of 34 nm (178 meV) and the *h*-BNCO-1-based device shows a green EL spectrum with a peak at 528 nm, an FWHM of 39 nm (173 meV) and a nearly Lambertian emission pattern (Supplementary Fig. 7a). With an appropriate EL peak as well as a narrow spectral bandwidth, the *h*-BNCO-1-based device finally operates at high-quality green CIE coordinates of (0.24, 0.71), which is very close to the CIE coordinates of (0.21, 0.71) in the green standard defined by the National Television System Committee (NTSC) (see Supplementary Fig. 7b). Moreover, the device based on *h*-BNCO-1 exhibits a maximum EQE of 40.1%, which is evidently superior to the BNCZ-based device (12.1%) and among the highest EQEs in OLEDs. Importantly, at an initial brightness of 1000 cd m$^{-2}$, the *h*-BNCO-1-based device still maintains a decent EQE of 34.6%, corresponding to a relative roll-off of 14%. Such efficiency roll-off behavior in the *h*-BNCO-based device is superior to not only the BNCZ-based device but also most of the ever-reported OLEDs exploiting binary EMLs with MR-emitters (see Supplementary Table 2). These device results further demonstrate the importance of *h*-BNCO in accelerating the RISC process and relieving exciton quenching at high current densities (see Supplementary Fig. 7c). The *h*-BNCO-1-based device simultaneously achieves high EQE, low-efficiency roll-off, and high color purity, which apparently surpasses the reported OLEDs.

Traditionally, the severe efficiency roll-off and poor operational stability of TADF-based devices could be mainly ascribed to triplet accumulation, which originates from the slow RISC process from the triplet to the singlet excited state. To further validate the benefits of the accelerated RISC process, operational EL stability based on *h*-BNCO-1 was subsequently evaluated. Herein, organic functional materials TAPC, TCTA, and TmPyPB with relatively fragile glass transition temperatures (79, 155, and 79 °C, respectively) were replaced by their more stable analogous N,N'-di(1-naphthyl)-N,N'-diphenyl-(1,1'-biphenyl)-4,4'-diamine (NPD, 95 °C), 9-phenyl-3,6-bis(9-phenyl-9Hcarbazol-3-yl)-9H-carbazole (TrisPCz, 163 °C), and 2-(9,9'-spirobi[fluoren]-

## Table 1 | Photophysical properties of BNCZ and *h*-BNCO-1

| Compound | $\lambda_{abs}$[a] [nm] | $\lambda_{em}$[a] [nm] | FWHM[b] [nm]/[eV] | $E_{S1}$[c] [eV] | $E_{T1}$[c] [eV] | $\Delta E_{S1T1}$[d] [eV] |
|---|---|---|---|---|---|---|
| BNCZ | 458 | 477 | 23/0.13 | 2.57 | 2.46 | 0.11 |
| *h*-BNCO-1 | 489 | 516 | 28/0.13 | 2.39 | 2.36 | 0.03 |

[a]Peak wavelengths of absorption and fluorescence spectra in toluene.
[b]Full-widths at half-maxima.
[c]Lowest excited singlet ($S_1$) and triplet ($T_1$) energy levels are determined from the peak values of fluorescence and phosphorescence spectra, respectively, in dilute toluene at 77 K.
[d]$\Delta E_{S1T1} = S_1 - T_1$.

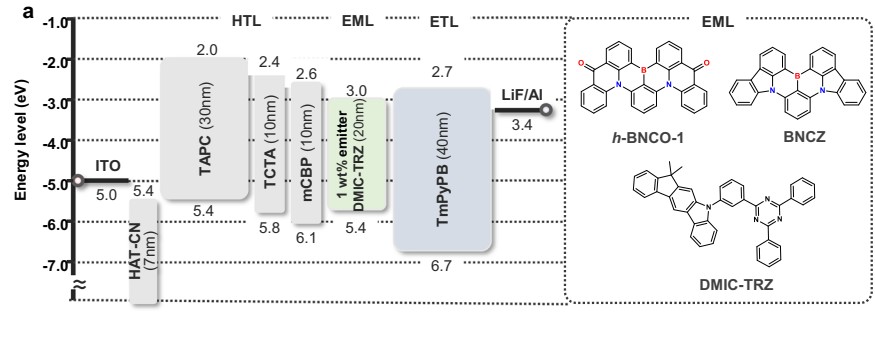

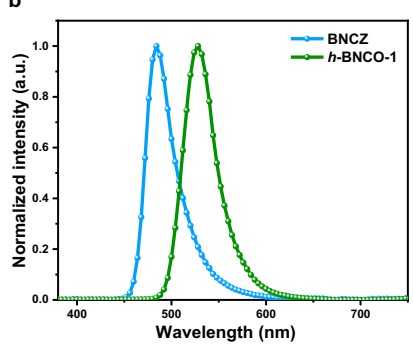

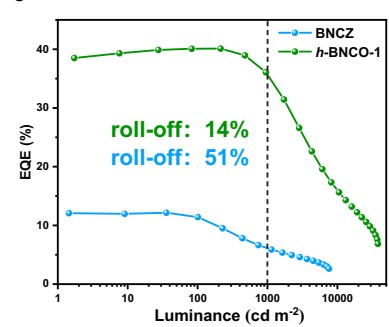

**Fig. 3 | Optimized EL performances of BNCZ- and *h*-BNCO-1-based OLEDs.**
**a** Device structures with ionization potential and electron affinity (in eV) for each material (HTL hole-transporting layer, EML emitting layer, ETL electron-transporting layer), and the relevant chemical structures of the materials used in the EMLs. **b** Normalized EL spectra. **c** EQE–luminance characteristics and efficiency roll-offs of the OLEDs, in which the black dashed line marks the luminance of 1000 cd m$^{-2}$, and the green and blue marks represent the device roll-off results at 1000 cd m$^{-2}$ of the devices based on *h*-BNCO-1 and BNCZ, respectively.

**Table 2 | EL performances of the OLEDs using DMIC-TRZ as the host**

| Emitters | λ$_{EL}$$^a$ [nm] | FWHM$^b$ [nm]/[meV] | EQE$_{max}$$^c$ [%] | EQE$_{1000}$$^d$ [%] | CE$_{max}$$^e$ [cd A$^{-1}$] | PE$_{max}$$^f$ [lm W$^{-1}$] | CIE $^g$ (x, y) |
|---|---|---|---|---|---|---|---|
| **BNCZ** | 484 | 35/182 | 12.1 (11.8 ± 0.3) | 5.96 | 22.6 | 23.6 | (0.12, 0.36) |
| ***h*-BNCO-1** | 528 | 39/173 | 40.1 (39.6 ± 0.5) | 34.6 | 159.2 | 147.0 | (0.24, 0.71) |

$^a$Peak wavelength of the EL spectrum.
$^b$Full-width at half-maximum.
$^c$The maximum external quantum efficiency, in which the average device parameters in parentheses are based on the measurement of over four independent devices.
$^d$The external quantum efficiency at 1000 cd m$^{-2}$.
$^e$The maximum current efficiency.
$^f$The maximum power efficiency.
$^g$CIE coordinates taken at EQE$_{max}$.

3-yl)-4,6-diphenyl-1,3,5-triazine (SF3-TRZ, 135 °C)[34]. In addition, the EML utilized 5,12-dihydro-12-(4,6-diphenyl-1,3,5-triazin-2-yl)-5-phenyl-indolo[3,2-a]carbazole (PIC-TRZ2)[36] with strong bonding energies as the host matrix with 1 wt% *h*-BNCO-1. As shown in Supplementary Fig. 8, the optimized device with a binary EML exhibited a low turn-on voltage of 2.5 V at a luminance of 1 cd m$^{-2}$, pure-green narrowband emission with CIE coordinates of (0.26, 0.70), and a high EQE of 25.0%. This efficiency is inferior to that of the device based on DMIC-TRZ and *h*-BNCO-1, mainly because $\Phi_{PL}$ of the EML is lower, *ca.* 77.8% ± 2%. On the other hand, benefitting from the high $k_{RISC}$ (>1 × 10$^5$ s$^{-1}$) of *h*-BNCO-1 and the device structure, the lifetime reaching 95% of the initial luminance (LT$_{95}$, starting from 1000 cd m$^{-2}$) was measured around 140 hours (Fig. 4a). Such device lifetime is 170 times longer than that of the control device with the identical device structure using BNCZ as the emitter (LT$_{95}$ = 0.81 h, starting from 1000 cd m$^{-2}$, see detailed in Supplementary Fig. 9). To our knowledge, this operational lifetime is one of the best performances in the reported MR-OLED based on a binary EML (Fig. 4b). Additionally, our device stability is also comparable to that of its ternary-EML analogous via extra assistance by TADF molecules or phosphors (see Supplementary Fig. 10 and Supplementary Table 3). Therefore, the impressive performance and

stability obtained via a simple binary-EML structure composed of only pure organics indicates the potential for future low-cost commercialization.

## Discussion
In conclusion, we proposed a promising molecular-design strategy—hybridization of organoboron-nitrogen and carbonyl (*h*-BNCO)—for MR emitters to achieve high color purity, high efficiencies, low-efficiency roll-off, and promising operational stability in OLED devices, which goes beyond the conventional ones. Compared to the control compound BNCZ, our designed *h*-BNCO-1 shows a similar narrow fluorescence-emission bandwidth of 0.13 eV and a higher $\Phi_{PL}$ (99.6% ± 2%). Importantly, *h*-BNCO-1 achieves a fast $k_{RISC}$ of 1.79 × 10$^5$ s$^{-1}$. This is mainly because the introduction of two carbonyl groups in *h*-BNCO-1 induces a smaller $\Delta E_{S1T1}$, a smaller $\Delta E_{T1T2}$, and a larger T$_2$-S$_1$ SOC value, thus facilitating the T$_2$-to-S$_1$ upconversion channel, which makes a large contribution to the overall RISC process. Ultimately, the optimized OLED based on binary EML with *h*-BNCO-1 achieves an ultrapure green EL with a spectral FWHM of 39 nm and CIE coordinates of (0.24, 0.71). Meanwhile, thanks to the enhanced RISC process, the device EQE not only achieves a maximum value of 40.1%

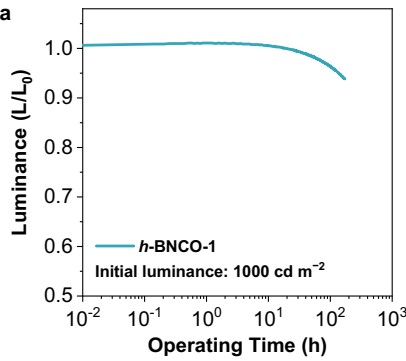

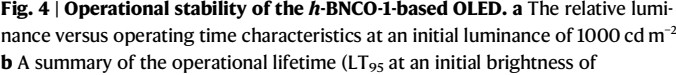

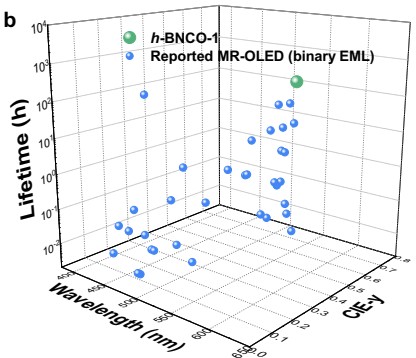

**Fig. 4 | Operational stability of the *h*-BNCO-1-based OLED. a** The relative luminance versus operating time characteristics at an initial luminance of 1000 cd m⁻². **b** A summary of the operational lifetime (LT$_{95}$ at an initial brightness of 1000 cd m⁻²)-wavelength peak-CIE-y coordinate of the reported MR-OLEDs based on binary-EMLs.

but also shows a much smoother efficiency roll-off compared to most of the device behaviors of the reported MR-TADF emitters. Moreover, *h*-BNCO-1 has demonstrated promising operational stability in OLEDs. In a device with a binary EML, a lifetime LT$_{95}$, as long as 137 h, is determined at an initial brightness of 1000 cd m⁻².

In the aspect of OLED material chemistry, our work provided a promising chemical paradigm of *h*-BNCO for designing more MR-TADF molecules; in the aspect of OLED devices, the impressive performance and stability obtained via a simple binary-EML structure with *h*-BNCO-1 demonstrated the potential for low-cost commercialization based on *h*-BNCO type MR-TADF molecules. While the EL color purity is slightly inferior to that of the previously reported purest green MR-TADF molecule[37], we believe it can be addressed by appropriately locking the phenyl-rings with the central phenyl-ring in *h*-BNCO-1 molecule (e.g., using spiro-carbon formation) to further suppress the excited-state geometrical distortion and vibration modes. Further molecular design and optimization are ongoing based on this concept.

## Methods
### Molecular synthesis and characterization
Current MR compounds with boron-fused polycyclic aromatics are synthesized via one-shot/one-pot borylation reactions based on an intramolecular tandem bora-Friedel-Crafts reaction, in which lithium-halogen exchange by using organic Li reagents (RLi) is generally carried out at the first step, and then a borylation reaction occurs via suitable boron trihalides. However, for a carbonyl-containing reaction substrate, on the one hand, RLi, as a highly active nucleophile, can easily react with carbonyl groups[38–43]; on the other hand, carbonyl groups substantially reduce the activity of aromatics for the Friedel-Crafts reaction. Therefore, *h*-BNCO frameworks have not been reported to the best of our knowledge. Herein, the synthesis route of our target *h*-BNCO framework *h*-BNCO-1 is displayed in Supplementary Fig. 11. Precursor 3 was first synthesized via the Buchwald–Hartwig reaction. Then, the preliminary B/N-based MR compound 4 containing a methylene group was synthesized by using 3 as the reaction substrate through a one-pot lithiation–borylation–annulation reaction. Finally, by using 4 as the intermediate, the target molecule *h*-BNCO-1 was obtained by further oxidizing the methylene for the carbonyl group. The detailed synthetic routes and characterization data, including NMR and high-resolution mass spectral (HRMS) analysis of all newly reported molecules, are shown in Supplementary Figs. 12–20 and the supplementary information. Moreover, the molecular geometry and packing mode of *h*-BNCO-1 in a single crystal sample were analyzed via X-ray diffraction (XRD) (see Supplementary Fig. 21, Supplementary Data 1, and Supplementary Tables 4 and 5).

## Material
All reagents were purchased from commercial sources and used without further purification. ¹H NMR and ¹³C NMR spectra were recorded on a Bruker 600/151 MHz spectrometer in deuterium reagent at room temperature. MALDI-TOF mass data were recorded on a Bruker ultrafleXtreme instrument.

## Computational details
To examine the equilibrium geometries of the ground and excited electronic states, the range-separated hybrid density functional ωB97XD with the nonempirical tuned ω value and the 6–31 G(d,p) basis set was employed here. All these quantum-chemistry calculations were performed using the Gaussian 16 software package[44]. Based on the optimized ground-state geometries, the high-level wavefunction-based STEOM-DLPNO-CCSD method with the basis set def2-TZVP implemented in ORCA 4.2.1 software[45] was exploited to examine the excited-state energies and difference-density plots of the studied molecules.

With the calculated electronic structure parameters in Supplementary Table 1 as the input, the rate constant ($k_{T_1 \to S_1}/k_{T_2 \to S_1}$) of the $T_1 \to S_1 / T_2 \to S_1$ process was evaluated via Marcus theory based on time-dependent perturbation theory:

$$k_{T_{1/2} \to S_1} = \frac{2\pi}{\hbar} SOC(T_{1/2} - S_1)^2 \frac{1}{\sqrt{4\pi\lambda_{T_{1/2}-S_1}k_B T}}$$
$$\exp\left(-\frac{(E_{S_1} - E_{T_{1/2}} + \lambda_{T_{1/2}-S_1})^2}{4k_B T\lambda_{T_{1/2}-S_1}}\right) \tag{1}$$

and the Boltzmann-averaged $k_{RISC}$ was evaluated via:

$$k_{RISC} = \frac{e^{-\frac{E_{T_1}}{k_B T}}}{e^{-\frac{E_{T_1}}{k_B T}} + e^{-\frac{E_{T_2}}{k_B T}}}k_{T_1 \to S_1} + \frac{e^{-\frac{E_{T_2}}{k_B T}}}{e^{-\frac{E_{T_1}}{k_B T}} + e^{-\frac{E_{T_2}}{k_B T}}}k_{T_2 \to S_1} \tag{2}$$

where $k_B$ is the Boltzmann constant; $T$ is the temperature; and $\lambda_{T_{1/2}-S_1}$ denotes the reorganization energy related to the transition from $T_1/T_2$ to $S_1$.

## Thermal stability
The thermal stability of BNCZ and *h*-BNCO-1 was evaluated via thermogravimetric analysis (TGA). As shown in Supplementary Fig. 22, the decomposition temperatures ($T_d$, corresponding to 5% weight loss) of BNCZ and *h*-BNCO-1 are estimated to be 359 °C and 383 °C, respectively. The even higher $T_d$ of *h*-BNCO-1 indicates that introducing

carbonyl groups further improves thermal stability, which is essential for the thermal evaporation process.

## Electrochemical measurement

Cyclic voltammetry (CV) measurements were carried out for BNCZ and *h*-BNCO-1 (Supplementary Fig. 23). Based on the onsets of the oxidation and reduction curves, the HOMO energy levels are estimated to be −5.61 and −5.83 eV, and the LUMO energy levels are estimated to be −2.94 and −3.29 eV for BNCZ and *h*-BNCO-1, respectively. With carbonyl incorporation in the polycyclic aromatics, LUMO energy levels are more substantially deepened than HOMOs. The energy gaps ($E_g$) of *h*-BNCO-1 (2.54 eV) are slightly narrower than that of BNCZ (2.67 eV), which would lead to a redshift emission.

## Photophysical measurements

UV–vis absorption spectra were recorded on a Hitachi U-3900 spectrophotometer. PL spectra were recorded on a Hitachi F-4600 fluorescence spectrophotometer. Transient fluorescence decays were measured with a time-resolved photoluminescence spectrometer (Edinburg FLS1000). The absolute PLQY was recorded on a Hamamatsu Quantaurus-QY quantum yield spectrometer (C13534-11).

## Molecular orientation measurement

We evaluated the molecular orientation of 1 wt% emitter doped in a DMIC-TRZ film via angle-dependent PL measurements. As shown in Supplementary Fig. 24, a much higher horizontal molecular orientation ratio ($\Theta_{//}$) of 98% is detected for *h*-BNCO-1 compared with 70% for BNCZ, which implies that devices based on *h*-BNCO-1 should exhibit higher light outcoupling efficiency than those based on BNCZ.

## Device characterization

The electrical characteristics of the devices were measured with a Keithley 2400 source meter. The EL spectra and luminance of the devices were obtained on a PR655 spectrometer. Note that we also prepared a device with 1 wt% BNCZ doped in 9-(2-(9-phenyl-9H-carbazol-3-yl)phenyl)-9H-3,9′-bicarbazole (PhCzBCz) film as an alternative EML to achieve better device efficiency (see Supplementary Fig. 25).

## Device operational lifetime measurement

The operational lifetime, driving voltage, luminance, and electroluminescent spectra of the encapsulated OLED devices were recorded and measured using a luminance meter (SR-3AR, TOPCON, Japan) under a constant current density with an initial luminance of 1000 cd m$^{-2}$.

## Data availability

The data that support the findings of this study are available in the supplementary material of this article. Crystallographic data for the structure reported in this article has been deposited at the Cambridge Crystallographic Data Center under deposition number CCDC 2224022; a copy of the data can be obtained free of charge via https://www.ccdc.cam.ac.uk/structures/. The new crystallographic structure of the target molecule is also available within the accompanying files. Source data are provided in this paper.

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

## Acknowledgements

This work was supported by the National Natural Science Foundation of China (Grant Nos. 52130304 (X.-H.Z.), 51821002 (X.-H.Z.), and 52003185 (K.W.)), the National Key Research & Development Program of China (Grant Nos. 2020YFA0714601 (K.W.) and 2020YFA0714604 (K.W.)), the Science and Technology Project of Suzhou (Grant No. ZXL2022490 (K.W.)), the Suzhou Key Laboratory of Functional Nano & Soft Materials, the Collaborative Innovation Center of Suzhou Nano Science & Technology, and the 111 Project (X.-H.Z.). X.T. and C.A. acknowledge the support of Japan Science and Technology Agency (JST) CREST (grant no. JPMJCR22B3). X.T. acknowledges support from JSPS KAKENHI (grant no. 22K20536). The authors thank Dr. Zhong Chen and Ms. Yuan Cheng from the Instrumentation and Service Center for Molecular Sciences at Westlake University for assistance with molecular orientation and characterization measurements and data interpretation.

## Author contributions

K.W., X.-K.C., C.A., and X.-H.Z. conceived and supervised the project. X.-K.C. and K.W. proposed the h-BNCO molecular design strategy. Y.-C.C., K.W., and X.-K.C. designed the h-BNCO compound. X.-K.C. carried out all of the theoretical simulations. Y.-C.C. synthesized and characterized the steady-state photophysics of the organic compound with the help of X.X. and X.-C.F. Y.-C.C. performed the photophysical measurements and fabricated the efficiency-optimized OLED devices. X.T. designed and fabricated the operational stability of OLED devices and characterized the transient PL curves. Y.-C.C., X.T., K.W., X.-K.C., C.A. and X.-H.Z. wrote the paper. X.-C.F., X.X., J.Y., S.L., R.W., Y.X., T.Z., and D.Z. commented on the paper. All authors discussed the results and commented on the final paper.

## Competing interests

The authors declare no competing interests.
