## [Peer Review File · Nature Communications]

Efficient, narrow-band, and stable electroluminescence from organoboron-nitrogen-carbonyl emitterEditorial Note: This manuscript has been previously reviewed at another journal that is not operating a transparent peer review scheme. This document only contains reviewer comments and rebuttal letters for versions considered at *Nature Communications*.

REVIEWER COMMENTS

Reviewer #1 (Remarks to the Author):

The authors have undertaken a thorough revision in response to the comments of all of the reviewers. I am glad to recommend this work for publication in Nature Communications at this stage.

Reviewer #2 (Remarks to the Author):

With this report, I am evaluating the revision NCOMMS-23-45430-T of a manuscript, which was carried out based on a manuscript transfer based on the original reviewers comments. In particular, I am assessing the revision related to the comments my review (Reviewer #2). Overall, I am not fully satisfied with the answers, which is why I relate here to the comments that I would like to see addressed again in another revision.

Reviewer 2, Comment 1:

With respect to the PLQY reporting, I have asked, what value of 100% means for the h-BNCO-1 molecule. Here, the authors acknowledge that this is not to indicate that the molecule has perfect PLQY, still they say it is very high. What I actually would want to see is the actual measurement value in conjunction with a reasonable error margin. In the revision, the authors have now moved from stating '100%' to 'nearly unity PLQY'. This is equally not scientific. As a general comment: I would like to learn about the actual error margin they found for their measurement system and have all PLQY values reported with this error included. Furthermore, the manuscript now contains a statement like (line 191) for another sample: '...evaluated to be 75% (within the experimental error)'. Same statements/approach in line 227. What is within the experimental error supposed to mean? Have the authors rounded the values? Please be more scientific and consistent.

Reviewer 2, Comment 4: Related to my comment on the misleading suggestion the abstract makes, the authors did not change the presentation. In the abstract one still finds that both high EQE and good stability are related to the emitter in an identical OLED structure. Here, the authors are clearly asked to specify that the two important performance metrics are not yet met in the same device architecture at once.

Reviewer #3 (Remarks to the Author):

Decision: "Recommend"

General comments: In this manuscript, Xiao-Hong Zhang et al., submitted a work on Efficient, narrow-band, and stable electroluminescence from organoboron nitrogen-carbonyl emitter (manuscript no. NCOMMS-23-45430-T). The authors have synthesized organoboron nitrogen-carbonyl-embedded molecules which enhanced the RISC rate compared with the earlier reports of BNCz molecule. Hence, they optimized the device and recorded more than 40% device efficiency and the operational stability LT95 137 h for 1000 cd/m². This manuscript is quite satisfactory and the molecule design concept can be an aid to the readers for the new molecular framework. Therefore, we recommend that the work be published in this journal.

Before that authors should revise the manuscript.

1. In the photophysical study, the authors mentioned toluene solution the FWHM remained unchanged compared with BnCZ molecules even though the emission red-shifted so we need some additional information authors provide the solvatochromic studies than we can understand your findings are good.
2. Why does the PLQY 75% seem to be low? The authors give reasons.
3. In Line no 245 authors described the quenching process have any evidence for this statement if anything else please provide the experimental proof.
4. In fig SI 8 clearly shows roll efficiency is very severe and why it readily decreases the EQEs. why?
5. The author's findings (h-BNCO-1) compared the design strategy and device performance of BNCz molecules well agreed. Even though previously reported Nat Commun 13, 4876 (2022) pure green emitters showed better performance. However, the proposed work and previous work compared have some lack in the molecule design strategy so please authors find that reason.

Point-to-Point responses to reviewers' comments

Reviewer #1: (Remarks to the Author):

The authors have undertaken a thorough revision in response to the comments of all of the reviewers. I am glad to recommend this work for publication in Nature Communications at this stage.

Our Response: We thank the reviewer for the satisfaction of our revised manuscript and the positive recommendation for publication.

Reviewer #2 (Remarks to the Author):

With this report, I am evaluating the revision NCOMMS-23-45430-T of a manuscript, which was carried out based on a manuscript transfer based on the original reviewers comments. In particular, I am assessing the revision related to the comments my review (Reviewer #2). Overall, I am not fully satisfied with the answers, which is why I relate here to the comments that I would like to see addressed again in another revision.

Our Response: We thank the reviewer for the thorough reading and suggestions. We have made additional efforts to address the reviewer's concerns and further improved the manuscript.

Reviewer 2, Comment 1:

With respect to the PLQY reporting, I have asked, what value of 100% means for the h-BNCO-1 molecule. Here, the authors acknowledge that this is not to indicate that the molecule has perfect PLQY, still they say it is very high. What I actually would want to see is the actual measurement value in conjunction with a reasonable error margin. In the revision, the authors have now moved from stating '100%' to 'nearly unity PLQY'. This

is equally not scientific. As a general comment: I would like to learn about the actual error margin they found for their measurement system and have all PLQY values reported with this error included. Furthermore, the manuscript now contains a statement like (line 191) for another sample: '...evaluated to be 75% (within the experimental error)'. Same statements/approach in line 227. What is within the experimental error supposed to mean? Have the authors rounded the values? Please be more scientific and consistent.

Our Response: We appreciate the reviewer's additional explanation of their concern and apologize for our unsatisfied revision in the previous round. The manufacturer's (Hamamatsu Photonics) calibration results indicate that the actual error margin for the PLQY measurement systems is $\pm 2\%$. The previous PLQY values were rounded, as the accuracy cannot be reached to one decimal place. In the revised manuscript, all the PLQY values have been restated as the actual measurement value \pm the actual error margin.

Reviewer 2, Comment 4: Related to my comment on the misleading suggestion the abstract makes, the authors did not change the presentation. In the abstract one still finds that both high EQE and good stability are related to the emitter in an identical OLED structure. Here, the authors are clearly asked to specify that the two important performance metrics are not yet met in the same device architecture at once.

Our Response: We are thankful for the reviewer's valuable suggestion and deeply sorry for our inappropriate description. The ambiguous/misleading descriptions in the abstract/introduction have been clarified to specify that the two key performance metrics have not yet been met simultaneously in the same device architecture.

Reviewer #3 (Remarks to the Author):

Decision: "Recommend"

General comments: In this manuscript, Xiao-Hong Zhang et al., submitted a work on Efficient, narrow-band, and stable electroluminescence from organoboron nitrogen-carbonyl emitter (manuscript no. NCOMMS-23-45430-T). The authors have synthesized organoboron nitrogen-carbonyl-embedded molecules which enhanced the RISC rate compared with the earlier reports of BNCz molecule. Hence, they optimized the device and recorded more than 40% device efficiency and the operational stability LT95 137 h for 1000 cd/m². This manuscript is quite satisfactory and the molecule design concept can be an aid to the readers for the new molecular framework. Therefore, we recommend that the work be published in this journal.

Before that authors should revise the manuscript.

Our Response: We thank the reviewer for recognizing the importance of our manuscript and positive recommendation.

1. In the photophysical study, the authors mentioned toluene solution the FWHM remained unchanged compared with BnCZ molecules even though the emission redshifted so we need some additional information authors provide the solvatochromic studies than we can understand your findings are good.

Our Response: We are thankful for the reviewer's valuable suggestion. We have carried out the solvatochromic measurement on both BNCZ and *h*-BNCO-1. As depicted in Supplementary Fig. 2b&c, as the solvent polarity increases, both emitters show similar emission behaviours, e.g., slight emission redshifts and mild spectral broadening. In each solvent, the spectral FWHMs of both emitters remained essentially unchanged in energy units, similar to those in toluene solution. The related description has been added to the revised manuscript.

2. Why does the PLQY 75% seem to be low? The authors give reasons.

Our Response: We are thankful for the reviewer's valuable suggestion. The relatively low PLQY of *h*-BNCO-1 doped in an mCBP film (75%) should be ascribed to the less sufficiently exciton transfer process between the host and dopant due to the less ideal overlap between the emission band of mCBP and the absorption band of *h*-BNCO-1. Similar phenomena were also reported in some previous literature (e.g., *Adv. Mater.* 35, 2300510 (2023); *ACS Appl. Mater. Interfaces* 13, 49076 (2021); *ACS Appl. Mater. Interfaces* 8, 22382 (2016)).

3. In Line no 245 authors described the quenching process have any evidence for this statement if anything else please provide the experimental proof.

Our Response: We are thankful for the reviewer's valuable suggestion. To support the exciton quenching process in the device, the EQE versus current density curve has been fitted using the triplet-triplet annihilation (TTA) and singlet-polariton annihilation (SPA) models. As depicted in Supplementary Fig. 7c, the actual curve aligns more closely with the TTA model at low current densities and with the SPA model at high current densities. These results support mixed exciton quenching processes occur and dominate the efficiency roll-off performance of the device.

4. In fig SI 8 clearly shows roll efficiency is very severe and why it readily decreases the EQEs. why?

Our Response: We are thankful for the reviewer's valuable suggestion. The EQE at 1000 cd m⁻² is 18.8%, corresponding to a relative efficiency roll-off of 24.8%, which is still a promising result among the reported MR-OLEDs based on a binary-EML. The efficiency

roll-off is more severe than that of the EL efficiency-optimized device mainly because of the less ideal carrier balance of the device, which is a necessary trade-off for improved device stability.

5. The author's findings (*h*-BNCO-1) compared the design strategy and device performance of BNCz molecules well agreed. Even though previously reported Nat Commun 13, 4876 (2022) pure green emitters showed better performance. However, the proposed work and previous work compared have some lack in the molecule design strategy so please authors find that reason.

Our Response: We are thankful for the reviewer's valuable suggestion. EL colour purity of our developed emitter *h*-BNCO-1 (e.g., FWHM=39 nm, CIE_y=0.71) is a bit inferior to that of the previously reported green emitter (e.g., FWHM=24 nm, CIE_y=0.75) in Nat Commun 13, 4876 (2022), which is a relative weakness of the overall performance. In terms of molecular design, this is mainly because the excited-state distortion and vibration modes are not fully suppressed. Inspired by the mentioned reference, it is expected that this issue can be further addressed by appropriately locking the phenyl-rings with the central phenyl-ring in the *h*-BNCO-1 molecule (e.g., using spiro-carbon formation). The mentioned reference has been cited and the related discussion has been added to the revised manuscript.

REVIEWERS' COMMENTS

Reviewer #2 (Remarks to the Author):

In the present 2nd revision NCOMMS-23-45430A, the authors have addressed all remaining points of my review in a satisfactory way. I am happy to recommend this work for publication.

Reviewer #3 (Remarks to the Author):

Current revised manuscript is good for publishing in Nature Communications. The authors addressed all the comments to be quite satisfactory.